# Psychotherapy with Psilocybin for Depression: Systematic Review

**DOI:** 10.3390/bs13040297

**Published:** 2023-03-31

**Authors:** Jonathan Joseph Dawood Hristova, Virtudes Pérez-Jover

**Affiliations:** Health Psychology Department, Miguel Hernández University, 03202 Elche, Spain; jonathan.dawood@goumh.umh.es

**Keywords:** psilocybin, depression, psychotherapy, review

## Abstract

Depression is a common mental health issue that affects 280 million people in the world with a high mortality rate, as well as being a leading cause of disability. Psychopharmacological therapies with psychedelics, particularly those with psilocybin, are showing promising potential for the treatment of depression, among other conditions. Some of their benefits include a rapid and exponential improvement in depressive symptoms and an increased sense of well-being that can last for months after the treatment, as well as a greater development of introspective capacity. The aim of this project was to provide experimental evidence about therapeutic procedures along with psilocybin for the treatment of major depressive disorder. The project highlights eight studies that examined this condition. Some of them dealt with treatment-resistant depression while others dealt with depression due to a life-threatening disease such as cancer. These publications affirm the efficiency of the psilocybin therapy for depression, with only one or two doses in conjunction with psychological support during the process.

## 1. Introduction

According to the World Health Organization [1], depression is a common illness, affecting approximately 280 million people worldwide. About 700,000 people with depression die by suicide each year, making it the second leading cause of death in young people aged 15 to 29 and a leading global cause of disability. Despite the existence of effective pharmacological therapies for depression, there is limited efficacy to this form of treatment. At times, it produces adverse effects and adherence problems in patients [2]. It has been predicted that 23% of patients with major depression will remit within 13 weeks without any treatment [3]. According to a study by Kolovos et al. [4], traditional treatments for depression have a remission rate of 33%, which is only 10% higher than those who remit without treatment. It is necessary to develop and investigate innovative and efficient alternative treatments after taking into account these factors and the considerable negative impact of this condition on public health [5].

Psilocybin is a natural tryptamine compound found in certain species of mushrooms. Its structure and mechanisms of action are similar to those of serotonin. Despite being classified as a Schedule I drug in the US, it is becoming popular again for therapeutic purposes, even though it has been used for thousands of years for healing and spiritual purposes. Clinical studies with psilocybin for depression treatment, among various treatment-resistant disorders, have yielded satisfactory results, increasing the amount of evidence over time and offering a promising paradigm for psychology and psychiatry [6,7].

## 2. Materials and Methods

### 2.1. Search Strategy

In April 2022, a literature search following the Preferred Reported Items for Systematic Reviews and Meta-Analysis declaration guide [8] was conducted to identify studies that evaluated psychological therapy combined with psilocybin for the treatment of depression. The search was carried out on databases such as ProQuest, PsycInfo, PubMed, ScienceDirect, and Scopus, using the keywords “psilocybin” and the name of each disorder, along with their derivatives and synonyms, using the Boolean operators “AND” and “OR” (“psilocybin” AND depress* AND [“therapy” OR “support” OR “treatment”]). The search was limited so that the keywords appeared in the title in all the databases. In addition, a manual search was carried out based on the bibliographic references of the selected publications. Data extracted from each study included the country, objective, condition, design, participants, treatment, dose, measurement instruments, results, and conclusions.

### 2.2. Inclusion and Exclusion Criteria

The inclusion criteria were based on research articles that focused on psychological treatment of depression supplemented with psilocybin. Those treatments were conducted in any population, in either English or Spanish, and without a date limit, as it is considered relevant to include all scientific evidence in a timeless manner, and without restrictions on the type of clinical trial. Documents such as theses, reports, book chapters, study protocols, descriptive articles, and reviews of other articles were excluded, as well as those that did not use the substance for therapeutic purposes and those based solely on pharmacological treatment. Exclusion criteria also included those that focused exclusively on spiritual experiences without measuring psychological variables, as well as those that primarily discussed the physiological effects of psilocybin on the brain.

### 2.3. Search Flow

Of the 88 studies that were identified during the search, 52 were selected after eliminating duplicates. After analyzing titles and abstracts, 11 articles were chosen for full-text reading. Additionally, one study was added based on the bibliographic references of the selected articles, resulting in a total of 12 articles selected for full-text reading, as the others did not meet the inclusion criteria. Following this, a full-text reading was conducted, and four more articles were excluded as they met exclusion criteria, leaving eight articles for extracting relevant information to perform the results’ analysis (see Figure 1).

## 3. Results

The most relevant data from the eight articles that met the inclusion criteria are presented below.

### 3.1. Objective of the Studies

All selected papers focus on psilocybin therapy to treat depressive disorder. However, certain differences should be mentioned regarding this condition, such as studies on patients who suffer from depressive symptoms due to their oncological situation [9,10] or in cases where the depressive disorder is treatment-resistant [11,12,13]. Furthermore, one of the included studies was based on the comparison of treatments between psilocybin and escitalopram [14].

### 3.2. Study Design

Of the eight studies analyzed, five had a randomized controlled design [2,9,10,14,15], with three of these five being double-blind [9,10,12] and two being crossover studies [9,10]. The remaining three studies were open-label trials [11,12,13].

### 3.3. Participants

The sample sizes ranged from 12 to 59 participants [9,14] with an age range of 39 to 56 years [9,15]. Regarding the sex of the patients, the number of males was higher than females in six out of eight studies, except for Davis et al. [2] and Gukasyan et al. [15]. The study by Carhart-Harris et al. [11] was the only one with an equal number of males and females (*n* = 6).

### 3.4. Treatment

It should be noted that certain studies share treatment due to their relationship, either by the same author [11,12,13] or to performing long-term follow-up [2,15].

The therapeutic processes carried out in these studies share several factors, beginning with preparatory sessions prior to dosing. The preparation period had a variable duration range, from just one session [11,12,13,14] to three or more sessions over several weeks [2,9,10,15]. The goal of the preparatory sessions is to discuss the participants’ history and the dosing procedure, but especially to create a therapeutic alliance. Following this procedure, two dosing sessions took place in all studies, which lasted between 6 and 8 h, in a cozy environment, in a reclined and comfortable position, with eyes closed and music therapy or with psychotherapy in the case of Ross et al. [9]. In studies with music therapy, therapists accompany patients during the process, adopting a non-directive and supportive attitude, allowing the patient to experience the inner experience uninterrupted. Generally, the doses had a one-week delay, except for the study by Giffiths et al. [10], in which they were delayed by five weeks, or Ross et al. [9], with a seven-week delay. Finally, after each dose, integration sessions were conducted, dedicated to listening to patients’ testimonies and occasionally providing some interpretation regarding the content of the experience and its potential meaning, in addition to helping process and develop a personal sense while maintaining positive aspects of it. The number of integration sessions varied between totals of two [11,12,13], four [2,15], or six [9,10].

The duration of the treatment phase varied between studies, although it was not specifically stated in some [11,12,13]; it ranged from 6 weeks [14] and 8 weeks [2,15] to several months [9].

A description of these issues is presented in Table 1.

### 3.5. Dosage

The range of psilocybin doses went from a moderate dose of 10 mg in the first sessions [11,12,13] to high doses of 30 mg/70 kg in the second sessions [2,15]. It is also worth noting that in double-blind studies, a small enough dose of psilocybin was used to act as a placebo, being 1 mg/70 kg [10] or 250 mg of niacin [9]. However, the most commonly used doses were 25 mg of psilocybin, in four of the eight studies [11,12,13].

### 3.6. Evaluation Measures

The evaluation measures used were quantitative in all studies, focusing solely on depressive symptoms. The most commonly used instruments were the Beck Depression Inventory (BDI) [9,10,11,12,13,15], Quick Inventory of Depressive Symptoms (QIDS) [2,11,13,14,15], and GRID–Hamilton Depression Rating Scale (GRID-HAMD) or its variant HAM-D [10,11,13,15]. The less common instruments were the Montgomery–Asberg Depression Rating Scale (MADRS) [13], Snaith–Hamilton Pleasure Scale (SHAPS) [13], and Prediction of Future Life Events (POFLE, 2006) [12]. The latter consists of predicting the possibility of certain desirable and undesirable everyday situations over a period of 30 days. After that time period, patients must indicate which situations have occurred.

### 3.7. Synthesis of the Results

The results of all studies showed a significant reduction in depressive symptoms after treatment with one or two doses of psilocybin. Symptomatic improvement was immediate in some cases, showing significant results one day [2,9] and one week after the second dose [2,11,12,13]. This improvement was long-lasting, maintaining significant reduction up to 6 [10,13], 8 [9], and 12 months [15]. Regarding the comparative treatment between psilocybin and escitalopram [14], no significant differences were found between both at the sixth week. Although secondary evaluation measures mostly favor psilocybin treatment, firm conclusions cannot be established as confidence intervals have not been adjusted in several comparisons. Nonetheless, both treatments were effective, and the study may have been underpowered to detect small differences.

On the other hand, moderate/high doses of psilocybin, compared to placebo or low doses of such a substance, showed considerably greater treatment efficiency. For example, in the study by Griffiths et al. [10], the group with a high amount of psilocybin in the first dose showed a much greater reduction in symptoms than the group with a low amount. However, when crossing the doses, the scores of the group with the low dose in the first place resembled the scores of the high-dose group. These observations were also demonstrated similarly in the study by Ross et al. [9], using niacin as a placebo instead of a low dose of psilocybin. These results demonstrate that a single moderate/high dose of psilocybin produces a significant improvement in symptoms.

In addition, all studies, except one [14], provided information regarding the magnitude of the differences, with effect sizes being large at all measurement points, favoring treatment with psilocybin. Finally, in six of the eight studies [2,9,10,11,14,15], adverse effects produced by psilocybin were mostly headaches, dizziness, nausea, and tachycardia, occurring occasionally and with moderate or mild intensity, not hindering treatment, and no cases of addiction to the substance were reported after treatment. It should also be noted that in the study by Carhart-Harris et al. [14], the adverse effects of the psilocybin and escitalopram groups were similar.

A description of these issues can be seen in Table 2.

## 4. Discussion

The objective of this work has been to present a review of therapies that have used psilocybin to treat depressive symptoms, as well as a synthesis of the effectiveness of this treatment. Eight studies have been found that focus on this treatment, with one of them being a comparison with escitalopram.

After analyzing the studies, the importance of the therapist’s role in the course of this substance is frequently emphasized. In the first place, building a therapeutic alliance is crucial for facilitating the initial stages of this experience The therapist must forge a sense of trust and closeness with the patient, which is necessary for them to feel comfortable and safe with undergoing the unfamiliar psilocybin therapy and to be sufficiently prepared for it. Unlike many other drugs, the use of this substance requires the therapists to accompany the patients throughout the transitory effect of this substance, as it leads them to an abstract and individual realm, making it important for the therapists to provide guidance and support throughout the experience. In addition, the therapists play a crucial role in helping the patients integrate the thoughts, feelings, and emotions stimulated by the experience. It is important to note that the setting in which the therapy is carried out is also significant, as it facilitates the revelation of the patients’ inner landscape. The ideal environment is a quiet space, in a reclined position, wearing an eye mask, and engaging in music therapy.

The effectiveness of the treatments is noteworthy, leading to optimal results in a few sessions. The fact that patients showed a rapid reduction in depressive symptoms as early as one day or one week after a dose highlights the difference compared to traditional treatments with other antidepressant drugs, which generally require at least two weeks to take effect with daily administration [16], and may take 6 to 12 months to produce optimal improvement [17]. Although the study by Carhart-Harris et al. [14] did not find significant differences between both drugs, the duration of the antidepressant effect of psilocybin up to several weeks with a single dose could facilitate the progress by avoiding daily medication. This might also improve the adherence to treatment, which is poor, with only around 50% [18] of patients treated with classical antidepressants, in addition to having limited efficacy, with 30–50% of patients not responding to treatment and 10–30% completely resistant to treatment [2].

Although it was not mentioned in the review, the influence of the spiritual experience is worth noting. This factor has been evaluated in more than half of the studies presented [9,10,11,12,15] and represents an important subjective component that contributes to attitudinal and behavioral improvement. That is, the intensity and quality of the psychedelic experience can act as a predictor of the maintenance of mental health benefits in the medium and long term [19,20].

The consumption of psilocybin or any psychoactive substance can produce psychotic symptoms such as delusions, panic attacks, and depersonalization [21,22]. However, when carried out in a controlled context, adverse effects are limited (headaches, dizziness, nausea, and tachycardia), as mentioned earlier. It is important to note that these adverse effects are mild or moderate and merely transient, unlike classical antidepressants, which mainly cause dizziness, gastrointestinal problems, and sexual dysfunction, among others [23,24], more continuously since such treatment involves repeated consumption. In addition, the studies presented in this work did not report any cases of potential abuse or addiction to psilocybin, being a substance with a low addictive risk and no evidence that it causes any withdrawal symptoms [2,25].

Regarding the limitations that have been presented in the development of this work, one of them is the scarcity of articles that evaluate the effectiveness of this treatment experimentally or in comparison with other traditional drugs. Additionally, it should also be noted that there was no access to scientific articles that would have been interesting for the review. Another limitation would be that, given the limited scope of this work, it was not possible to include articles that examine the treatment of other psychological disorders. Nevertheless, a satisfactory analysis of a significant part of this field has been carried out. It should be noted that half of the studies have been carried out in the United States [2,9,10,15] and the other half in the United Kingdom [11,12,13,14], which leads to the conclusion that there is a lack of population representativeness. Two other factors to consider regarding the study sample would be the imbalance between men and women, as there has been a great disparity in the present studies, and the participants’ history of substance use, as having experience with psychedelics or other drugs could bias adaptation to the treatment.

There is a certain homogeneity among the authors in this research area, with Carhart-Harris and Griffiths being the main researchers and inspirers of most of the studies with psilocybin. The combination of this factor with limited experimental support means that the number of studies conducted on this topic is relatively small, with many of the studies using similar or related experimental conditions. It would be convenient to carry out a broader reflection on procedural differences, such as variability in doses, treatment, or a greater number of studies with a double-blind and controlled design, in order to have more experimental rigor. Regarding the study design, an important factor to consider is the influence of the type of preparation prior to the doses, as certain expectations could bias the procedure when combined with a placebo, facilitating the identification of the substance and thus hindering the double-blind design. In addition, another aspect that should be controlled in future clinical trials is the effect of expectancy masking. These elements should be taken into account and be measured routinely, as discussed in the work of Muthukumaraswamy et al. [26].

Future research in this area could explore the po31tential benefits of other types of more extensive therapeutic interventions, such as cognitive-behavioral or acceptance and commitment therapy [27]. Given that most of the studies presented in this review used music therapy in conjunction with minimal therapeutic intervention (mainly non-directive psychological support), it would be valuable to investigate the influence of different types of psychotherapy of the effectiveness on this modality. Likewise, investigating which psychotherapies would be more functional in relation to high or low doses of psilocybin would help expand its effectiveness and extension to other sociocultural and economic levels.

As mentioned earlier, the spiritual factor could be of utmost importance in terms of patients’ experience and improvement during the treatment. It would be interesting to explore this topic and have more solid references, so that it is easier for professionals to understand and guide patients during their spiritual experiences.

Other important factors to consider, apart from the focus of this review, could be the extrapolation of psilocybin therapy for the treatment of other disorders. So far, research has been found experimenting with its efficacy with anxiety, a factor that is valued in some articles presented in this review [9,10] and addictions, such as alcoholism [28,29,30] or smoking [31,32,33], as well as speculations regarding its possible effectiveness for obsessive-compulsive disorder [34] and post-traumatic stress disorder [35,36,37], among others. The inclusion of different psychotherapies evidenced for these disorders and the development of procedural guides and structures [29,38] to prepare and adapt professionals to these types of therapies could represent a revolutionary expansion of therapeutic approaches.

Finally, although there is limited evidence of the use of psilocybin in non-clinical populations, recent studies have suggested its potential benefits. For example, research has shown that microdosing in healthy individuals can have positive effects on personality traits and yield promising results. Given this, exploring the use of microdoses in this population may be a worthwhile avenue for further research [39,40,41].

## 5. Conclusions

In conclusion, psilocybin treatment for depression represents a promising paradigm for the fields of psychology and psychiatry. The growing number of experimental studies that demonstrate the efficiency of this substance highlights its therapeutic potential and minimizes adverse effects. Therefore, even though psilocybin is still classified as a harmful substance due to its legal and cultural history it could lead to a positive revolution in this field and become a novel antidepressant intervention. By carrying out a procedurally appropriate and adaptive use, it could significantly expand the range of possible medical applications, such as depression, post-traumatic stress disorder, addictions, and obsessive-compulsive disorder.

## Figures and Tables

**Figure 1 behavsci-13-00297-f001:**
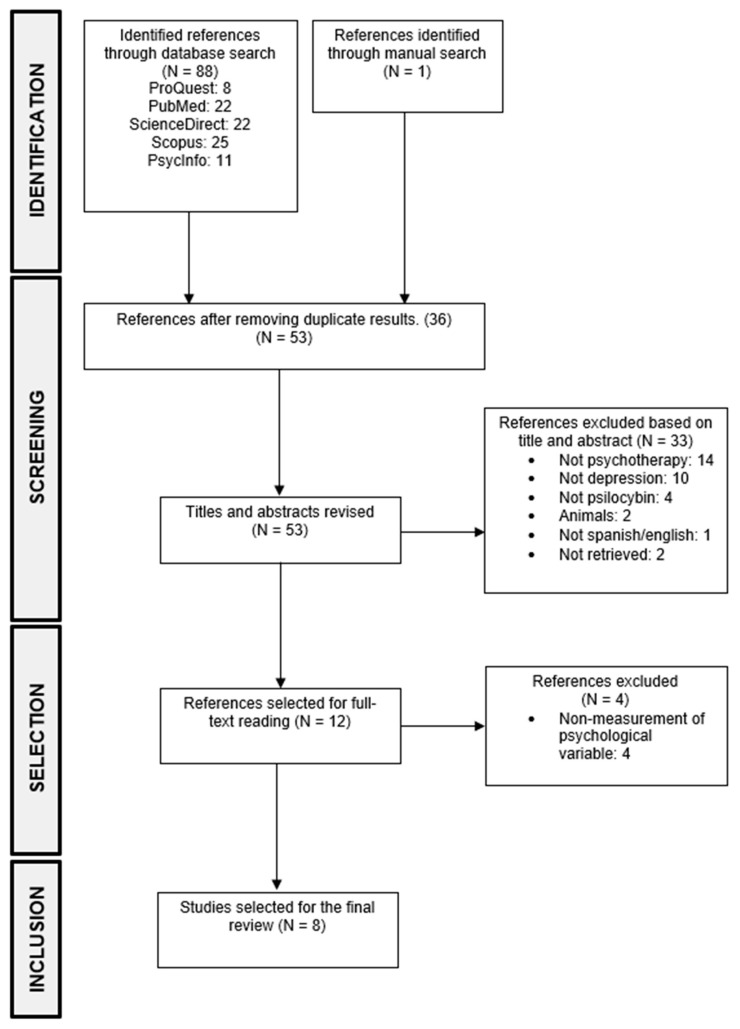
PRISMA flow diagram of the study selection.

**Table 1 behavsci-13-00297-t001:** Authors, objectives, conditions, designs, participants, and treatments of the studies.

Author	Objective	Condition ^1^	Design	Participants	Treatment
Ross et al., 2016 [9]	To study the effects of psilocybin therapy on cancer patients with depressive and anxious symptoms.	D&A	Double-blind, controlled, randomized, crossover	- Group 1 (first placebo/second psilocybin): N = 15 (27% men, 73% women). Mean age: 60.27 - Group 2 (first psilocybin/second placebo): N = 14 (50% men, 50% women). Mean age: 52 - Total N = 29 (18 men, 11 women). Mean age: 56.28	Three preparatory sessions (4 weeks) to establish therapeutic alliance. Then, two dosing sessions (alternating psilocybin or placebo) with psychotherapy separated by 7 weeks, supported by therapists with a non-directive and supportive attitude. Between doses, three integration sessions (total of 6 h) were conducted. Finally, three additional integrative sessions (total of 6 h) were conducted after the second dose, over a period of 6 weeks.
Carhart-Harris et al., 2016 [11]	To study the feasibility, safety, and efficacy of psilocybin in patients with major depressive disorder.	MDD (TR)	Open-label	N = 12 (6 men, 6 women). Mean age: 42.7	One preparatory session (4 h), followed by two doses with music therapy, separated by one week. After each dose, an integrative session was conducted.
Giffiths et al., 2016 [10]	To study the effects of psilocybin therapy on cancer patients with depressive and anxious symptoms.	D&A	Double-blind, controlled, randomized, crossover	- Group 1 (first high dose/second low dose): N = 25 (52% men, 48% women). Mean age: 56.1 - Group 2 (first low dose/second high dose): N = 26 (50% men, 50% women). Mean age: 56.5 - Total N = 51 (51% men, 49% women). Mean age: 56.3	Three preparatory sessions, followed by two doses of psilocybin (alternating high and low) separated by 5 weeks, and six sessions for dialogue and analysis of the experience with psilocybin (three between doses and three after the second dose).
Lyons & Carhart-Harris, 2018 [12]	To investigate the effects of a psilocybin intervention on pessimistic biases in patients with treatment-resistant depression.	MDD (TR)	Open-label, controlled trial (mixed model)	- Control group (healthy): N = 15 (60% men, 40% women). Mean age: 37.6 - Experimental group (depressed): N = 15 (73.3% men, 26.7% women). Mean age: 45.4 - Total N = 30 (66.7% men, 33.3% women). Mean age: 41.5	One preparatory session (4 h), followed by two doses with music therapy, separated by one week. After each dose, an integrative session was conducted.
Carhart-Harris et al., 2018 [13]	To study the efficacy and safety of psilocybin therapy for depression over a period of 6 months.	MDD (TR)	Open-label	N = 20 (14 men, 6 women). Mean age: 42.2	One preparatory session (4 h), followed by two doses with music therapy, separated by one week. After each dose, an integrative session was conducted.
Carhart-Harris et al., 2021 [14]	To compare psilocybin treatment with escitalopram treatment.	D&A	Double-blind, controlled randomized	- Escitalopram group: N = 29 (20 men, 9 women). Mean age: 39.1 - Psilocybin group: N = 30 (19 men, 11 women). Mean age: 43.3 - Total N = 59 (39 men, 20 women). Mean age: 41.2	Six weeks of treatment with six sessions: the first one for preparation before the first dose; the second one for taking the first dose; the third one for exploring the experience during the first dose; the fourth one for taking the second dose; the fifth one for psychological integration of the doses; and the sixth one for exploring the treatment experience.
Davis et al., 2021 [2]	To investigate the effect of psilocybin therapy in patients with major depressive disorder.	D&A	Controlled, randomized	- First group (immediate treatment): N = 13 (5 men, 9 women). Mean age: 43.6 - Second group (delayed treatment): N = 11 (4 men, 7 women). Mean age: 35.2 - Total N = 24 (8 men, 16 women). Mean age: 39.8	Eight weeks of intervention: three weeks of preparatory sessions for the doses (8 h total), two sessions with doses in weeks 3 and 4 of treatment along with music therapy, and two integration sessions (2–3 h) after each dose. One group received immediate treatment with psilocybin, and the other received treatment after 8 weeks (at the end of the first group), to differentiate the psilocybin intervention from spontaneous improvement.
Gukasyan et al., 2022 [15]	To study the efficacy and safety of psilocybin therapy over 12 months.	D&A	Controlled, randomized	- First group (immediate treatment): N = 13 (5 men, 9 women). Mean age: 43.6 - Second group (delayed treatment): N = 11 (4 men, 7 women). Mean age: 35.2 - Total N = 24 (8 men, 16 women). Mean age: 39.8	Eight weeks of intervention: three weeks of preparatory sessions for the doses (8 h in total), two sessions with doses in weeks 3 and 4 of treatment along with music therapy, and two integration sessions (2–3 h) after each dose. One group underwent immediate treatment with psilocybin, and the other group underwent treatment after 8 weeks (at the end of the first group’s treatment), to differentiate psilocybin intervention from spontaneous improvement.

^1^ MDD: major depressive disorder; TR = treatment resistant; D&A = depression and anxiety.

**Table 2 behavsci-13-00297-t002:** Dosage, measurement instruments, results, and conclusions of the studies.

Author	Dosage	Measurement Instruments ^1^	Results	Conclusion
Ross et al., 2016 [9]	0.3 mg/kg psilocybin or 250 mg niacin (placebo)	- HADS-D - BDI	Group 2 had a more significant reduction in symptoms than Group 1 in HADS-D scores 1 day, 2, 6, and 7 weeks after the first dose (d = 1.23 (*p* ≤ 0.001), d = 1.12 (*p* ≤ 0.01), d = 1.32 (*p* ≤ 0.001), and d = 0.98 (*p* ≤ 0.001), respectively). The results in the BDI measure were similar (d = 1.10 (*p* ≤ 0.01), d = 0.99 (*p* ≤ 0.01), d = 1.07 (*p* ≤ 0.01), and d = 0.82 (*p* ≤ 0.05)). Group 1 showed significant results after crossing over to the second dose for up to 8 months, with a symptom remission rate of 60% in BDI and HADS-D. Group 2 had a more stable symptom remission rate throughout the treatment, ultimately reaching 80% in both instruments.	Psychotherapy and a moderate dose of psilocybin produce a rapid and prolonged antidepressant response, which can last from 7 weeks to 8 months. The administration of psilocybin with proper psychotherapy could become a novel treatment for cancer-related depression.
Carhart-Harris et al., 2016 [11]	Dose 1: 10 mg psilocybin; Dose 2: 25 mg psilocybin	- HAM-D - QIDS - BDI - MADRS	The QIDS scores showed a significant decrease in symptoms from baseline (M = 19.2 (2.0)) with a maximum reduction at 2 weeks (M = 6.3 (4.6); g = 3.2, *p* = 0.002) and up to 3 months (M = 10.0 (6.0); g = 2.0, *p* = 0.003) after the last dose. The scores also decreased considerably one week after the last dose in BDI (33.7 (7.1) vs. 8.7 (8.4); g = 3.2, *p* = 0.002), in HAM-D (21.4 (4.5) vs. 7.4 (6.9); g = 2.4, *p* = 0.003) and in MADRS (31 (5) vs. 9.7 (9.8); g = 2.7, *p* = 0.002).	Psilocybin has a novel pharmacological action compared to current depression treatments. Hence, it could be a useful adjunct to therapies for this disorder.
Giffiths et al., 2016 [10]	High dose of psilocybin: 22 mg/70 kg; Low dose of psilocybin (placebo): 1 mg/70 kg	- GRID - GRID-HAMD-17 - BDI	The scores of the GRID-HAMD-17 measure reflected a significant reduction in group 1 after the first dose compared to baseline (22.84 (0.97) vs. 6.64 (1.04); *p* < 0.001), unlike group 2 (22.32 (0.88) vs. 14.80 (1.45)). These scores were similar after the crossover in the second dose, remaining constant for up to 6 months (6.95 (1.24), *p* < 0.001 in group 1 and 6.23 (1.30); *p* < 0.001 in group 2, with a combined effect size of d = 2.98). The results of the BDI instrument after the first dose also favored group 1 (17.77 (1.61) vs. 7 (1.39); *p* < 0.01) compared to group 2 (18.40 (1.09) vs. 12.92 (1.58) with a combined effect size), remaining similar at 6 months with a combined effect size of d = 1.63 (*p* < 0.001).	Therapy with a single dose of psilocybin produces a substantial and enduring decrease in depressive mood, at least up to 6 months.
Lyons & Carhart-Harris, 2018 [12]	Dose 1: 10 mg psilocybin; Dose 2: 25 mg psilocybin	- BDI, 1961) - POFLE	There was a significant reduction in BDI scores in the experimental group from baseline to 1 week after the last dose (M = 34.33 (7.44) vs. 12.13 (9.8); g = 1.9, *p* < 0.001) compared to the control group (M = 3.67 (3.83) vs. 2.73 (3.41); *p* = 0.284). According to the results of the POFLE measure, the experimental group made a similar prediction of desirable and undesirable future events (M = 0.29 (0.15) vs. M = 0.23 (0.15); *p* = 0.317, respectively), but ultimately had more desirable than undesirable events (M = 4.6 (1.76)) vs. M = 1.4 (1.35); g = 1.5, *p* < 0.001). The control group predictions of desirable and undesirable events (M = 0.57 (0.09) vs. M = 0.22 (0.16); g = 2.0, *p* < 0.001) were more realistic, ultimately resulting in more desirable than undesirable events (M = 5.6 (1.45) vs. M = 2.23 (1.84); g = 1.2, *p* < 0.001).	The findings suggest that psilocybin therapy may decrease pessimistic traits and give patients a more accurate and realistic perspective of their future.
Carhart-Harris et al., 2018 [13]	Dose 1: 10 mg psilocybin; Dose 2: 25 mg psilocybin	- QIDS-SR16 - BDI - SHAPS - HAM-D	QIDS-SR16 scores were significantly reduced at 1, 2, 3, 5 weeks, and 3 months post-treatment (d = 2.2, 2.2, 2.1, 2.3, and 1.5 (*p* < 0.001), respectively). HAM-D scores had a significant reduction in scores at 1 week post-treatment (24.1 (5.4) vs. 9.3 (7.6); d = 2.3, *p* < 0.001). BDI scores were significantly reduced at 1 week, 3 months, and 6 months (d = 2.5, 1.4, and 1.4 (*p* < 0.001), respectively). SHAPS scores were also significantly reduced at 1 week post-treatment (d = 1.3, *p* < 0.001) and at 3 months (d = 0.8, *p* < 0.005).	Two sessions with psilocybin produced a rapid and significant improvement in depressive symptoms, which was maintained for up to 6 months post-treatment. This substance represents a promising paradigm for treatment-resistant depression.
Carhart-Harris et al., 2021 [14]	25 or 1 mg of psilocybin in the dosing sessions. Daily capsule consumption, placebo for the psilocybin group and 10 mg of escitalopram for the comparison group. After the second dosing session, the escitalopram consumption was doubled (20 mg) for 3 weeks.	- QIDS	In the sixth week of treatment, there were no significant differences in QIDS scores between the psilocybin and escitalopram groups (−8.0 (1.0) vs. −6.0 (1.0), *p* = 0.17, respectively). In that week, 21 patients (70%) in the psilocybin group and 14 patients (48%) in the escitalopram group showed a treatment response (without a significant difference). Remission of symptoms was also observed in 17 patients (57%) in the psilocybin group and 8 (28%) in the escitalopram group.	According to the QIDS-SR-16 measure, this trial did not show a significant difference in antidepressant effects between psilocybin and escitalopram.
Davis et al., 2021 [2]	Dose 1 (moderate-high): 20 mg/70 kg psilocybin; Dose 2 (high): 30 mg/70 kg psilocybin	- GRID-HAMD - QIDS	The scores of the GRID-HAMD measure in the first group significantly improved from baseline (22.9 (3.6)) to 1 week (8.0 (7.1); d = 2.3, *p* < 0.001) and 4 weeks (8.5 (5.7); d = 2.3, *p* < 0.001) after the second dose. The scores of the QIDS measure also had a significant decrease from baseline to 1 day after the first dose (16.7 (3.5) vs. 6.3 (4.4); d = 2.6, *p* < 0.001) and remained until 4 weeks after the second dose (6.0 (5.7); d = 2.3, *p* < 0.001). The scores of the second group remained constant before their respective treatment.	The results demonstrate that psilocybin therapy is a fast, sustainable way to treat major depressive disorder.
Gukasyan et al., 2022 [15]	Dose 1 (moderate-high): 20 mg/70 kg psilocybin; Dose 2 (high): 30 mg/70 kg psilocybin	- GRID-HAMD - QIDS - BDI- II	The mean scores on GRID-HAMD in the overall sample gradually decreased at 3, 6, and 12 months post-treatment, all showing significant differences (*p* < 0.001) with very large effect sizes (d = 2.0, 2.6, and 2.4, respectively). The rates of symptom remission on the QIDS measure were generally consistent at these time points (58%, 67%, and 67%, respectively), as were those on the BDI-II (58%, 75%, and 75%, respectively).	The psilocybin-assisted treatment for depression produced considerable antidepressant effects up to 12 months post-treatment.

^1^ HADS-D: hospital anxiety and depression scale—depression; BDI: Beck depression inventory; BDI-II: Beck depression inventory II; HAM-D: 21-item Hamilton depression rating scale; QIDS: 16-item quick inventory of depressive symptoms; MADRS: Montgomery–Asberg depression rating scale; GRID-HAMD-17: GRID–Hamilton rating scale for depression; POFLE: prediction of future life events; QIDS-SR16: 16-item quick inventory of depressive symptoms—self-report, SHAPS: Snaith–Hamilton pleasure scale.

## Data Availability

Data is available from the corresponding author.

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
