# Peer review of "Psychotherapy with Psilocybin for Depression: Systematic Review"

_behavsci, 2023, doi:10.3390/bs13040297_

Round 1

Reviewer 1 Report

This systematic review examines the literature for psilocybin for the treatment of depression. This is a hot topic currently about a potentially promising treatment, and it is absolutely true that we need better new treatments to improve the limited outcomes of existing treatments. It is good to see that they potentially included papers in two languages rather than just English.

The paper is presented well overall. The discussion is well done although the last sentence is perhaps overstating the evidence to date. Another point that could be made is about the negative emotion generated in the placebo arm once people recognise that they don’t have the active arm, which exaggerates the effect size in the active arm (e.g. see 10.1080/17512433.2021.1933434).

However, I have some issues with the paper however in its current form.

There are already quite a few systematic reviews so the authors need to justify the need for this review i.e. what is that gap in the literature that this paper is addressing?

My major issue concerns the rigour of the methods.  The authors do not refer to having registered the review in Prospero or a similar registry, do not refer to having followed a recommended guideline like PRISMA, or doing quality assessments of the included studies. These omissions are significant as those processes provide methodological rigour to give the reader confidence that the review methods are transparent and replicable. Related to this:

·        A limited number of databases were used

·        That seems like a very low number of papers identified in the original search. For example, I replicated their search with the terms provided – over 1000 papers were located including reviews, animal studies etc

·       The description of the search terms is insufficient to replicate this search.

·        In the limitations, there was suggestion that inability to source references was a problem related to the low number of included papers. If so, incomplete capturing of the literature may significantly limit the utility of this review. Inclusion and exclusion criteria need to be stated.

If the research was more rigorous than currently described in the manuscript, that search should be included as a supplementary document.

Minor issues

·        Abstract- perhaps “leading” cause of disability might be better than “first”

·        References- Ref 17- please check capitalisation

·        P 4 missing line between the first and second references

·        The content of the tables could be potentially be written more concisely e.g.  you can use abbreviations for the measures, with the full name being spelled out in table notes. Bullet points are clearer for lists etc.

·        In table 2, the study number (or preferably same info as column 1 in the previous table) needs to be included in the first column

·        P9 considerable effect size would be better stated as very large effect sizes.

·       Section 3 doesn’t have a title. That should probably be the results with the current results title being replaced by something like Synthesis of the results.

·       The pagination is not continuous

·       Section 3.1  Please reword the sentence lines 6-8 - as written, it sounds like that paper was included for other reasons, rather than meeting inclusion criteria

·       3.4. It is important that the same participants are not counted twice. Where studies use the same participants but report different data ( e.g. follow-up data), please cross-reference those studies for the reader

·       3.4 line 38- what does The longevity of the period refer to? Duration of the treatment phase?

·       3.6 line 61 and earlier. There was no significant difference between psilocybin and escitalopram however it is important to state that both were effective. That study would be underpowered to detect those differences.

·       4 second page  line 107. “Which is scarce at around 50%” needs rephrasing. Are the authors talking about low adherence to antidepressants/ treatment?

·       126 Do you think that no-one reportedly abused psilocybin later might be due to careful screening of participants in those studies i.e. absence of substance-related disorders?

·       Section 4 line111 – the paragraph discussing spirituality makes a good point- would be a good recommendation for future research

Reviewer 2 Report

The aim of the paper "Psychotherapy with psilocybin for depression: systematic review" is to register experimental evidence about therapeutic procedures with psilocybin for the treatment of major depressive disorders.

By analyzing eight studies, the authors concluded that psilocybin treatment for depression represents a truly promising agent minimizing adverse effects, despite being classified as a harmful substance.

1. A native English speaker should check the entire Manuscript.

2. Title: The title is adequate.

3. Abstract: The abstract is adequately addressed. However, regarding the sentence „Psychopharmacological therapies with psychedelics, especially with psilocybin, are showing a promising potential for the treatment of depression, among other conditions”, the authors should briefly state what are the benefits of psilocybin usage.

4. Introduction: The introduction is adequately written, however, there are some issues that should be addressed. “On the other hand, psychedelics are a class of hallucinogenic drugs that produce cognitive alterations and distortions in reality”. “On the other hand” should be dismissed since it is redundant. Additionally, a little bit more about psilocybin should be written, since there is only one sentence where psilocybin is mentioned in the Introduction. For example, its origin, production, structure and classification, mechanism of action and impact of its usage… could be unknown to the readers.

5. The Materials & Methods: This section is well described. Nonetheless, it would be nice to include in the Manuscript (in the “2.2. Search flow” section) what the exclusion criteria were (that were already stated in the Figure 1 "PRISMA flow diagram of the study selection"- not psychotherapy, not depression, not psilocybin...)

6. Results: The results are clearly presented and adequately addressed and the selected articles are recently published. The authors never mentioned any of the Tables in the text (for example: “It can be seen in Table 1” or “As presented in Table 2”…). Also, it should be mentioned in more detail what escitalopram is since it is mentioned several times.

7. Discussion: The discussion is well-written and adequately addressed. Unfortunately, as the authors already stated in the Manuscript, there is a scarcity of articles that evaluate the effectiveness of psilocybin treatment experimentally or in comparison with other traditional drugs. Furthermore, there is a certain homogeneity among the authors in this research area. Thus, it would be nice if the authors of this particular Manuscript, could provide (in the future, not in this Manuscript) some experimental work which could prove psilocybin efficacy.

Overall, the paper "Psychotherapy with psilocybin for depression: systematic review" is written in an appropriate manner, it is well-structured and interesting to read. This study is technically sound. Although the study does not contribute novel knowledge by itself, it is likely to attract a wide readership, therefore, the Manuscript can be considered for the publication.
